# Obstacles and facilitators of parents' coping with hematopoietic stem cell transplantation of a child with cancer

Batool Pouraboli[1], Maryam Maleki[2]*, Nahid Dehghan Nayeri[3], Amir Ali Hamidieh[4], Abbas Mardani[5]

**1** Department of Pediatric and Neonatal Intensive Care Nursing Education, School of Nursing and Midwifery, Tehran University of Medical Sciences, Tehran, Iran, **2** Social Determinants of Health Research Center, Research Institute for Prevention of Non-Communicable Diseases, Qazvin University of Medical Sciences, Qazvin, Iran, **3** Nursing and Midwifery Care Research Center, School of Nursing and Midwifery, Tehran University of Medical Sciences, Tehran, Iran, **4** Pediatric Cell and Gene Therapy Research Centre, Gene, Cell & Tissue Research Institute, Tehran University of Medical Sciences, Tehran, Iran, **5** Non-communicable Diseases Research Center, Research Institute for Prevention of Non-Communicable Diseases, Qazvin University of Medical Sciences, Qazvin, Iran

* malekimaryamn92@gmail.com

## Abstract

### Background

Pediatric hematopoietic stem cell transplantation (HSCT) is a complex process that impacts the entire family. The traumatic nature of the pediatric HSCT period makes this a particularly vulnerable time for parents, leading to coping challenges. This study aimed to explore parents' experiences regarding the obstacles and facilitators of coping with their child's HSCT.

### Methods

This qualitative study used a conventional content analysis method. The study took place at largest Children's Medical Center in Iran from February to November 2023. The study utilized purposive sampling for selecting participants. Data collection began with unstructured interviews, followed by in-depth semi-structured interviews with open-ended questions. Sampling continued until data saturation was achieved after examining qualitative data from 20 participants.

### Results

The qualitative analysis identified eight subcategories grouped into two main categories: "variable support" and "beliefs and individual situation". Support varied widely, with significant roles played by family, friends, healthcare providers, non-governmental organizations, and desirable beliefs and individual situation. However,

**Data availability statement:** All relevant data are within the manuscript.

**Funding:** The author(s) received no specific funding for this work.

**Competing interests:** The authors have declared that no competing interests exist.

inadequate family support, financial stress, and conflicts with healthcare teams were notable barriers.

## Conclusions

The findings underscore the need for comprehensive support systems and targeted interventions to address the emotional and practical challenges families face during their child's HSCT vulnerable period. Future efforts should focus on enhancing support structures and addressing barriers to improve the overall coping experience for parents.

---

## 1. Introduction

Each year, about 400,000 children worldwide are diagnosed with cancer, with nearly 70% under the age of 15 [1,2]. In Iran, childhood cancer ranks sixth in Asia for incidence, with cases increasing from 129.2 per million in 1999 to 132.13 per million in 2016 [3,4]. Over the past four decades, advances in pediatric cancer treatment have markedly improved survival rates, with many childhood cancers now achieving a 5-year survival rate exceeding 80% [5,6]. Among these advances, hematopoietic stem cell transplantation (HSCT) has become a crucial therapeutic option for both malignant and non-malignant conditions [7,8]. Despite its potential to save lives, HSCT is an intensive and complex procedure associated with serious medical risks, long recovery times, and profound psychosocial challenges for patients and their families [9,10].

The HSCT journey places parents under considerable strain. Even before transplantation, they face the emotional impact of the diagnosis and uncertainty regarding treatment outcomes [11]. Following HSCT, they must cope with complications, adhere to strict medical regimens, manage hygiene to prevent infections, monitor for transplant rejection, and provide continuous daily care after discharge [12,13]. These responsibilities create substantial emotional and practical burdens throughout both the pre- and post-HSCT phases [14,15]. Studies show that families of children undergoing HSCT experience higher stress levels than those managing other chronic childhood illnesses [16,17], affecting social relationships, financial stability, and coping abilities [18]. Previous research has documented elevated rates of anxiety, depression, distress, and post-traumatic stress among parents during this period [19–22], along with sleep disruption and reduced overall mental health.

Given the highly stressful nature of HSCT [23], effective coping strategies are essential to maintain parents' psychological well-being and the stability of family life [10,24]. These strategies can mitigate the intense stress and anxiety that commonly accompany the treatment process [10,24]. The present study draws on Folkman and Lazarus's Transactional Model of Stress and Coping, which conceptualizes coping as a dynamic interaction between individuals and their environment [25]. In this model, stress responses are shaped first by primary appraisal—evaluating the significance and potential threat of an event—and then by secondary appraisal—assessing

personal resources, abilities, and options for responding [26,27]. Identifying the obstacles and facilitators that influence these appraisals is key to understanding how parents manage the demands of their child's HSCT.

While several studies have examined parental coping in pediatric HSCT internationally, little is known about these experiences in the Iranian context, where sociocultural norms, family structures, and healthcare resources may uniquely shape the coping process. A clearer understanding of these factors can guide healthcare providers in delivering culturally sensitive and comprehensive support [28]. Therefore, this qualitative study aimed to explore parents' perspectives on the obstacles and facilitators to coping with their child's HSCT, providing insights to inform targeted interventions and improve parental quality of life.

## 2. Methods and materials

### 2.1. Design and participants

This qualitative study employed a conventional content analysis approach, a method used to systematically code and interpret textual data [29]. The study was conducted at the Children's Medical Center, affiliated with Tehran University of Medical Sciences, from February to November 2023. As the largest pediatric HSCT facility in Iran, this center performs approximately 100 pediatric HSCTs annually.

The inclusion criteria for participants were as follows: parents of children undergoing HSCT for cancer, fluency in Persian, the ability to articulate their experiences, and willingness to participate. Exclusion criteria included parents with a diagnosed mental illness, parents whose child was receiving HSCT for a non-malignant condition, or parents who had experienced the loss of a child during the HSCT process. The article followed the principles outlined in the Standards for Reporting Qualitative Research (SRQR), as detailed in Supplementary File 1.

### 2.2. Ethical considerations

The study protocol received approval from the Ethics Committee of Tehran University of Medical Sciences (Approval Number: IR.TUMS.CHMC.REC.1401.145). Participants were informed about the study's purpose, confidentiality of data, voluntary participation, anonymity, and their right to withdraw from the study at any point. Written informed consent was obtained from participants before each interview, with participants agreeing to audio recording as part of the study process.

### 2.3. Data collection

Participants were selected using purposive sampling with the explicit aim of achieving maximum variation in demographic and clinical characteristics, such as parental gender, age, education level, occupation, economic status, number of children, and the child's age, gender, diagnosis, and type of HSCT. To ensure diversity while minimizing potential gatekeeping by clinical staff, nurses were asked to introduce the study broadly to all eligible parents without screening based on perceived willingness or ability to participate. The second author (MM) then independently reviewed medical records to purposively select additional participants from varied socioeconomic backgrounds, geographic locations, and transplant types (autologous and allogeneic) until the desired variation was achieved.

Recruitment began with parents present at the hospital during follow-up visits or hospitalization, followed by contacting eligible parents via telephone using contact details provided in medical records, with permission from the hospital's ethics committee. Data collection began with unstructured interviews, progressing to in-depth semi-structured interviews guided by open-ended questions. The interviews were conducted following the interview guideline. Sample questions from the interview guide included: [1] "Can you describe the factors that help you cope with your child's HSCT?" and [2] "What are the factors that make coping with your child's HSCT more difficult?" Probing questions such as "Can you elaborate on that?" and "Can you provide an example?" were used to deepen the interviews (Supplementary file 2). Interviews lasted

between 40–60 minutes. A sociodemographic questionnaire was also used to gather information about the participants' age, education, economic status, number of children, occupation, place of residence, and details about the child, such as age, gender, birth order, type of cancer, type of HSCT, and time elapsed since HSCT.

The final sample size of 20 participants was determined according to data saturation, defined operationally as the point at which no new codes, categories, or themes emerged in three consecutive interviews and the codebook demonstrated stability. This approach ensured that thematic redundancy had been reached, consistent with qualitative research standards [30,31].

The research team comprised one PhD candidate in nursing with professional experience in caring for children undergoing HSCT (female), three nursing faculty members with extensive expertise in qualitative research, and one pediatric HSCT subspecialist physician. All team members had prior training in qualitative methods. The PhD candidate, who had established rapport with clinical staff but no prior personal relationship with participants, conducted all interviews. Her professional background facilitated understanding of the HSCT context, while awareness of potential bias prompted careful use of open-ended questioning and active listening to avoid leading responses. Reflexive notes were kept after each interview to document the interviewer's assumptions, emotional reactions, and potential influences on the data collection process. Throughout the study, regular team discussions were held to critically examine how researchers' positions and experiences might shape data interpretation, ensuring that findings were grounded in participants' perspectives.

### 2.4. Data analysis

Interviews were transcribed verbatim by the second author (MM). The data were analyzed using the conventional content analysis method, strictly following the steps outlined by Graneheim and Lundman [30,31]. Transcripts were read repeatedly to gain a comprehensive understanding of the content. Meaning units were identified and condensed, then coded manually in Microsoft Word without the use of specialized software. The coding process was conducted independently by two researchers (MM and AM), and any discrepancies were resolved through discussion; if consensus was not reached, a third author was consulted. Codes were then inductively grouped into subcategories and categories based on similarities and differences, ensuring that the analytical process remained consistent with conventional content analysis rather than thematic analysis [29,31].

### 2.5. Trustworthiness of data

The study followed Lincoln and Guba's criteria to ensure trustworthiness, addressing credibility, dependability, transferability, confirmability, and authenticity [32]. To enhance credibility, we employed prolonged engagement with participants, maximum variation in participant selection, and reflexivity. Member checking was conducted by sharing preliminary findings with several participants to verify the accuracy and resonance of the interpretations. Peer validation involved two external qualitative researchers reviewing the coding and categorization process. Triangulation was achieved by comparing data from multiple participants with diverse demographic and clinical characteristics, and by involving multiple researchers in data analysis. Dependability was ensured through clear documentation of research objectives, methodological steps, and decision-making processes. Transferability was supported by providing detailed descriptions of participants, the study setting, and contextual factors. Confirmability was strengthened by maintaining an audit trail of coding decisions, analysis notes, and reflexive memos, ensuring that findings were grounded in the data. Authenticity was maintained through transparent communication with participants, obtaining informed consent, and respecting participants' perspectives throughout the study.

## 3. Results

### 3.1. Demographic characteristics

The study included a diverse group of 20 participants, comprising 8 fathers and 12 mothers. Participants' ages ranged from 26 to 53 years, with educational backgrounds varying significantly, from no formal education to academic degrees.

Economically, they represented a wide range of income levels. Most participants had two children (n = 12). Regarding their children, the majority were girls and typically first-born, with ages ranging from 4 to 21 years. A significant portion of the children (n = 11) were diagnosed with Acute Lymphoblastic Leukemia (ALL), and 13 underwent allogeneic HSCT, while 7 underwent autologous HSCT. The time elapsed since HSCT ranged from 2 to 123 months. Detailed sociodemographic information is provided in Table 1.

**Table 1. Demographic profile of the participants.**

| Parent's sociodemographic data | | | | | | Child undergoing HSCT | | | | | |
|---|---|---|---|---|---|---|---|---|---|---|---|
| Participant | Age (year) | Education level | Number of children | Occupation | Economic status (self-report) | Age (year (month)) | Gender | Birth rank | Type of cancer | Type of HSCT | Time passed from HSCT (years (months)) |
| 1-Mother | 31 | Diploma | 2 | House-keeper | Not-sufficient | 7(6) | Girl | 2nd | WT[1] | Autologous | 1(2) |
| 2-Father | 36 | Diploma | 2 | Employed | Not-sufficient | 7(6) | Girl | 2nd | WT | Autologous | 1(2) |
| 3-Mother | 48 | Diploma | 2 | House-keeper | Relatively sufficient | 15 | Girl | 2nd | ALL[2] | Allogeneic | 7 |
| 4-Mother | 43 | Associate degree | 2 | House-keeper | Not-sufficient | 6 | Girl | 2nd | AML[3] | Allogeneic | 18 |
| 5-Father | 33 | Bachelor's degree | 2 | Employed | Sufficient | 5 (6) | Girl | 1st | ALL | Allogeneic | 11 |
| 6-Mother | 36 | Bachelor's degree | 2 | Employed | Sufficient | 10 | Boy | 1st | ALL | Allogeneic | 14 |
| 7-Father | 37 | Diploma | 1 | Self-employment | Not-sufficient | 9 | Boy | 1st | ALL | Allogeneic | 5 (7) |
| 8-Father | 47 | Bachelor's degree | 1 | Employed | Relatively sufficient | 14 | Boy | 1st | ALL | Allogeneic | 54 |
| 9-Father | 37 | Associate degree | 2 | Employed | Relatively sufficient | 9 | Boy | 1st | NB[4] | Autologous | 3 |
| 10-Mother | 39 | Under diploma | 1 | House-keeper | Not-sufficient | 13 | Girl | 1st | NB | Autologous | 54 |
| 11-Mother | 32 | Diploma | 2 | House-keeper | Not-sufficient | 5 | Girl | 2nd | ALL | Allogeneic | 4 (20) |
| 12-Father | 42 | Bachelor's degree | 2 | Self-employment | Not-sufficient | 5 | Girl | 2nd | ALL | Allogeneic | 4 (20) |
| 13-Mother | 38 | Bachelor's degree | 1 | House-keeper | Not-sufficient | 1 | Boy | 1st | NB | Autologous | 50 |
| 14-Mother | 40 | Associate degree | 2 | House-keeper | Not-sufficient | 5 (10) | Girl | 2nd | WT | Autologous | 5 (15) |
| 15-Father | 41 | Bachelor's degree | 3 | Self-employment | Not-sufficient | 18 | Boy | 1st | ALL | Allogeneic | 3 (5) |
| 16-Mother | 26 | Under diploma | 1 | House-keeper | Not-sufficient | 4 (8) | Girl | 1st | NB | Autologous | 6 [5] |
| 17-Mother | 38 | Illiterate | 3 | House-keeper | Not-sufficient | 14 | Girl | 1st | ALL | Allogeneic | 3 |
| 18-Mother | 36 | Bachelor's degree | 2 | Employed | Relatively sufficient | 7 | Girl | 2nd | ALL | Allogeneic | 5 (3) |
| 19-Father | 38 | Under diploma | 5 | Worker | Not-sufficient | 11 | Boy | 3th | AML | Allogeneic | 72 |
| 20-Mother | 53 | Under diploma | 4 | House-keeper | Relatively sufficient | 21 | Boy | 4th | ALL | Allogeneic | 123 |

## 3.2. Qualitative findings

Through comprehensive data analysis, we identified 8 subcategories, which were grouped into 2 main categories. The detailed results are presented in Table 2 and visually illustrated in Fig 1.

### 3.2.1 Variable support.
Participants experienced varying levels of support from family members, acquaintances, healthcare professionals, the healthcare system, and non-governmental organizations. Adequate support helped parents cope with their child's HSCT, while inadequate support impeded their coping efforts. The category of variable support encompasses the following subcategories: "support from family and friends", "insufficient support from family and friends", "support from the healthcare system", "challenges in support from the healthcare system", and "support from non-governmental organizations".

### 3.2.1.1. Support from family and friends.
Family and friends of most participants provided essential financial, instrumental, and emotional support, significantly easing the management of their child's HSCT. Spouses, in particular, offered extensive emotional and practical help, including taking care of other children, offering comfort, managing necessities, feeding the child undergoing HSCT, maintaining a constant presence, and supplying necessary drugs and equipment.

*"When I learned about my daughter's illness and her need for a transplant, my husband was my source of comfort and support. He was my constant solace and the only consolation I had."* (Participant 16, mother)

Participants also received crucial support from family and friends, who provided encouragement, empathy, and solidarity. This included shaving their heads in solidarity with the child, staying with the donor child, visiting, praying, and making offerings for recovery. Financial assistance from family and friends helped cover international donor costs, medicines, transportation, and housing during the HSCT period. Families and communities also assisted in various ways, such as taking care of other children, providing accommodation, helping with education, supplying equipment during quarantine, and finding platelet donors.

*"My sisters and brother were always there for us, especially my sisters who provided vital emotional support. They respected infection precautions and stayed in touch regularly."* (Participant 20, mother)

Support from peers, particularly other parents of children undergoing HSCT, was also vital. This included sharing information about the HSCT process, providing comfort, and offering encouragement.

*"Other parents of children undergoing HSCT shared valuable information with us about the stages of transplantation and treatment."* (Participant 15, father)

**Table 2. Subcategories and categories developed in this study.**

| Subcategory | Category |
|---|---|
| Support from family and friends | **Variable support** |
| Insufficient support from family and friends | |
| Support from the healthcare system | |
| Challenges in support from the healthcare system | |
| Support from non-governmental organizations | |
| Facilitating individual beliefs | **Beliefs and individual situation** |
| Facilitating individual situation | |
| Barriers related to individual circumstances | |

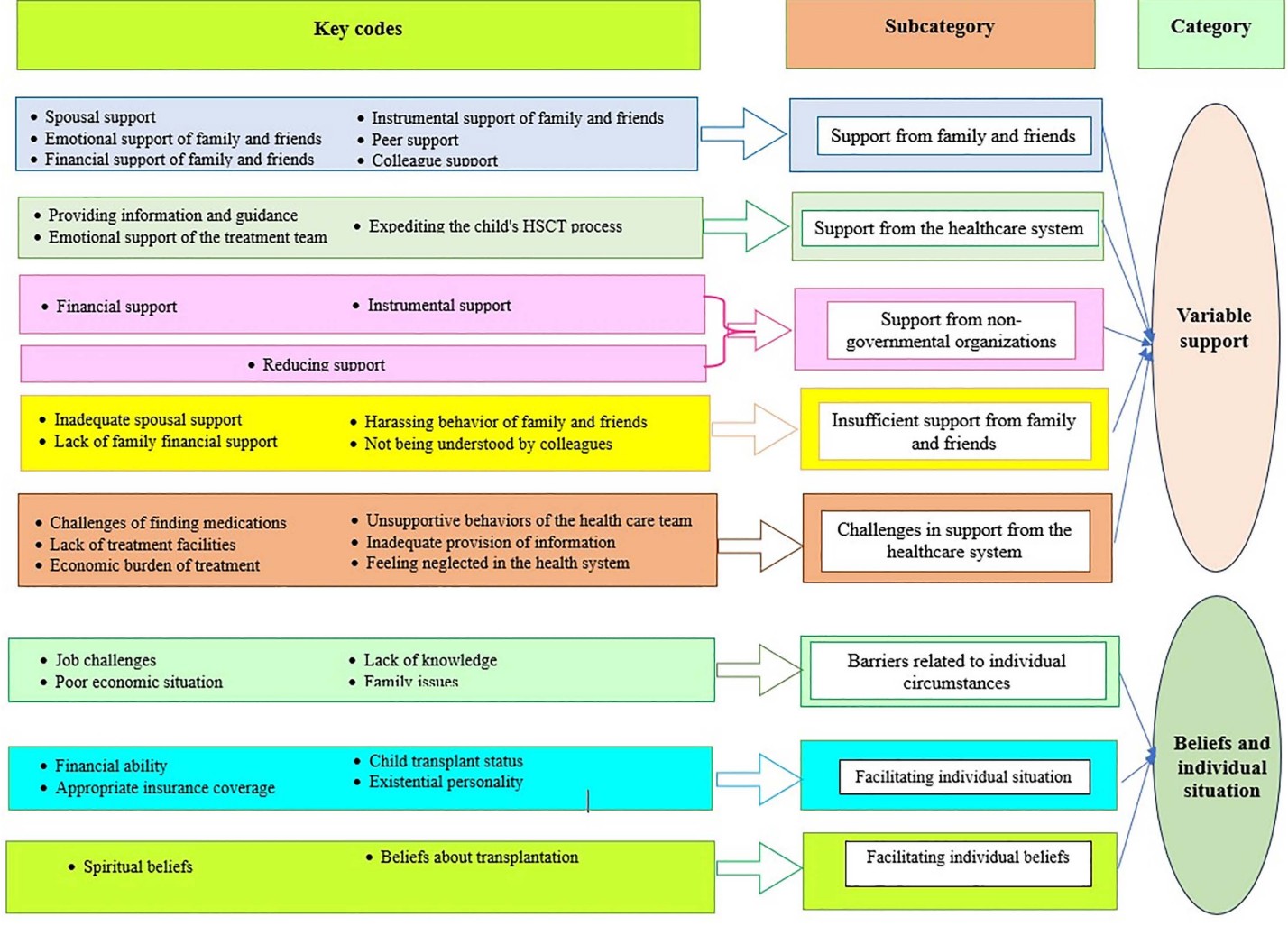

**Fig 1. The summary of the categories, subcategories and key codes.**

Some participants also emphasized the crucial support they received from colleagues during their child's illness and HSCT. This support included financial aid, exemptions from military service, provision of leave, and cooperation from colleagues. Additionally, colleagues' relatives donated platelets, and workplace agreements facilitated transfers to the city where participants lived. Colleagues also helped by providing fuel for their cars.

*"Some colleagues provided crucial support, including financial help and fuel cards, without expecting repayment."* (Participant 9, father)

**3.2.1.2. Insufficient support from family and friends.** Despite receiving support, some participants faced challenges with insufficient assistance from family and friends during their child's HSCT journey, which hindered their ability to cope. Issues with spousal support were notably distressing for some participants.

*"My husband didn't help with taking our daughter to the doctor or getting her tests done. For instance, I attended most of the pre-transplant medical consultations alone because he wouldn't come."* (Participant 4, mother)

Financial support from family also posed challenges for some, particularly in cases of poverty. Participants expressed frustration over the lack of financial assistance from their families.

*"I did not receive any financial support from my own family or my husband's family because they were all poor and couldn't help us."* (Participant 16, mother)

Additionally, some participants experienced distressing behavior from family and friends, including lack of sympathy, being blamed, and inappropriate comments.

*"One of my husband's brothers was inconsiderate, entering our house despite health risks, which caused stress and frustration."* (Participant 1, mother)

Support from colleagues was also lacking for some participants. Complaints included inadequate financial help, lack of understanding of their situation, and negative attitudes from coworkers.

*"Colleagues failed to understand our financial and work situation, and some behaved poorly, making things more difficult."* (Participant 15, father)

**3.2.1.3. Support from the healthcare system.** The healthcare system and providers offered valuable support to parents of children undergoing HSCT, which facilitated their coping. Most participants reported receiving comprehensive information and training from the healthcare team on various aspects of their child's treatment and HSCT. This included details about the illness and treatment options, HSCT procedures and potential complications, communication and dietary restrictions for the child, follow-up tests and visits, medication administration, health protocols, infection prevention, medical procedures such as port and CV line insertion, and referrals to charitable organizations and pharmacies. Participants expressed satisfaction with this support.

*"Nurses explained the port line's purpose to reduce frequent IV access and gave detailed hygiene and medication instructions after the transplant."* (Participant 8, father)

Healthcare providers also offered emotional support through various means, including arranging consultations with psychologists, providing comfort and encouragement, visiting the child, and offering reassurance.

*"The transplant doctor encouraged us to stay positive and personally visited my daughter in both the provincial center and the hotel, which was very comforting."* (Participant 18, mother)

Additionally, participants noted that the healthcare system facilitated the HSCT process through measures such as providing transplant medications, finding platelet donors, conducting searches for suitable international donors, coordinating with international donors, prioritizing the child for transplantation, securing medical currency from the Ministry of Health to cover the costs of international donors, and implementing an efficient appointment scheduling system.

*"When no platelet donors were available in Tehran, the doctor reassured us and the hospital quickly arranged one to avoid delays."* (Participant 16, mother)

**3.2.1.4. Challenges in support from the healthcare system.** Participants faced several challenges related to their child's HSCT, which affected their ability to cope effectively. Many participants struggled with the scarcity of essential medications. They reported significant distress and frustration due to the frequent unavailability of drugs and being sent between different organizations for support.

*"I felt hopeless from constantly hearing "We don't have it," and expressed my despair through poetry."* (Participant 12, father)

The absence of necessary medical facilities in their hometown required participants to travel long distances for treatment, which added to their difficulties.

*"Our city didn't even have a PET scan facility for my daughter, let alone the capability to perform a transplant. That's why we had to travel a long distance, which was very difficult."* (Participant 5, father)

The high costs associated with HSCT, including medications, tests, and travel, placed significant financial strain on families. Some participants had to borrow money or sell personal assets to cover these expenses.

*"The cost of some of my daughter's medications was high, and we always had to borrow money because we had to pay for some of her medications out of pocket."* (Participant 17, mother)

Participants experienced distress from unsupportive behavior by healthcare professionals, such as pessimistic communication and lack of direct engagement.

*"One of the doctors at the hospital told us that our son might not live for more than a month or 50 days... He spoke very hopelessly."* (Participant 20, mother)

Participants felt they were not given sufficient information about the treatment process, potential side effects, and medication management.

*"The doctor provided only brief explanations of the chemotherapy process, mentioning cell removal, freezing, and reinfusion."* (Participant 14, mother)

Issues such as difficulties with admissions, incorrect diagnoses, and errors in medication were cited as examples of negligence within the healthcare system.

*"During the follow-up visits after the transplant, it's like I have to directly approach the doctor myself... It makes you feel neglected in the system."* (Participant 9, father)

**3.2.1.5. Support from non-governmental organizations.** The MAHAK Institute, a non-governmental charitable organization supporting children with cancer, provided significant instrumental and financial assistance that greatly aided participants in managing their child's HSCT. Their support included covering the costs of chemotherapy, travel, accommodation, and medications, which helped alleviate the financial burden and logistical challenges faced by families.

*"Without MAHAK's support covering chemotherapy, travel, and accommodation costs, we could not have managed."* (Participant 19, father)

*"MAHAK covered many of my daughter's medication costs. They also paid for her hospitalization expenses."* (Participant 11, mother)

Despite this support, some participants experienced a reduction in assistance from MAHAK, which impacted their ability to cope.

*"Now, MAHAK no longer covers the cost of medications we buy out-of-pocket or the CT scans and MIBG tests every three months."* (Participant 16, mother)

**3.2.2. Beliefs and individual situation.** Parents' ability to cope with their child's HSCT is influenced by various individual factors. An analysis of participants' statements identified beliefs and personal circumstances that can either facilitate or hinder coping. These factors are categorized into three areas: "facilitating individual beliefs", "facilitating individual situation", and "barriers related to individual circumstances". Each category is discussed in detail below.

**3.2.2.1. Facilitating individual beliefs.** Participants identified various beliefs that influenced their ability to cope with the challenges of their child's HSCT, often facilitating the coping process. One prominent belief was the role of spiritual and religious faith. Many participants saw their child's illness and HSCT as a divine test or part of a greater plan, believing that with God's help, they could endure and adapt to the difficulties. Some even attributed positive events to divine intervention.

*"My child's illness and transplant were God's will; perhaps there was wisdom in it…"* (Participant 20, Mother)

*"Let me put it this way: God was testing us with our child's illness and transplant... God helped us in many situations, and I felt His assistance in our lives..."* (Participant 12, Father)

Another belief that facilitated coping was viewing the transplant as a hopeful and transformative event. Participants saw it as the final treatment option, a new beginning, or a significant step towards ending their child's suffering and repeated hospitalizations. This perspective helped them manage the challenges associated with HSCT more effectively.

*"I saw the transplant as a rebirth for my daughter and the end of her painful treatments and hospitalizations."* (Participant 1, Mother)

**3.2.2.2. Facilitating individual situation.** Interviews revealed that certain factors could influence participants' ability to cope with the challenges posed by their child's HSCT and facilitate their coping process. Some participants mentioned financial stability as a facilitating factor. Others highlighted having health insurance and supplemental insurance that covered hospital and medication expenses for their child as essential for their coping with their child's HSCT.

*"Thank God, our financial situation was good. I was even able to provide financial support to my own family. At least I could handle many of the treatment expenses."* (Participant 5, Father)

The status of the child's transplant was another factor identified as facilitating the parents' coping process. Some parents of children who underwent autologous HSCT noted that the autologous nature of the transplant made coping easier due to fewer complications, a quicker process, and no need for donor compatibility testing. Conversely, parents whose children received allogeneic HSCT found it easier to cope if a family member, such as a father or sibling, could be a donor, or if the mother was pregnant and could use the new baby's cord blood cells for the transplant.

*"The doctor mentioned that his father and sister could also be donors, which made me feel that the situation was a bit better."* (Participant 3, Mother)

**3.2.2.3. Barriers related to individual circumstances.** Barriers related to individual circumstances identified in the interviews were significant barriers to parental coping with their child's HSCT. Several participants noted that having a weak, anxious, introverted, or fearful personality negatively impacted their ability to cope.

*"I've always had anxiety and been a very anxious person. I get stressed and anxious over every little thing, so coping with my child's condition was quite difficult for me."* (Participant 6, mother)

Challenges in maintaining job activities also adversely affected coping. Participants described how difficulties during their child's HSCT period led to job loss and unemployment.

*"Job loss and financial difficulties caused significant mental strain, making it impossible to continue working."* (Participant 15, father)

Poverty and poor economic conditions further hindered coping. Many participants spoke about their dire financial situations.

*"We have no house, no vehicle, nothing. We're below zero. We even sold all the gold we had. Now we have no home, nothing... We couldn't even afford the ticket to come to Tehran."* (Participant 10, mother)

Another significant barrier was the participants' lack of awareness. Insufficient information and awareness about their child's illness, HSCT, dietary restrictions during HSCT, necessary tests, the cell harvesting process, CV line, port, BMA, chemotherapy, medication side effects, visits to MAHAK Institute, and psychiatric consultations were noted.

*"We had no prior information about the transplant... We weren't familiar with the transplant process at all. We knew nothing about the port they were going to place for our child."* (Participant 8, father)

Family issues, including divorce, poor relationships with spouses, and family conflicts, also emerged as obstacles.

*"my husband didn't have a good relationship with me, so I couldn't go to him for support. We had marital conflicts, and those problems were there too."* (Participant 10, mother)

## 4 Discussion

The present study aimed to elucidate parents' experiences of barriers and facilitators in coping with their child's HSCT through a qualitative content analysis. child's HSCT through a qualitative content analysis. Drawing on the Transactional Model of Stress and Coping, our findings can be interpreted as dynamic processes of primary and secondary appraisal. Parents first engaged in primary appraisal by perceiving HSCT as a threatening event, while secondary appraisal involved evaluating the resources available to manage these demands. Family support, financial means, health system support, and spiritual beliefs acted as resources that strengthened secondary appraisal and enabled adaptive coping. In contrast, limited resources such as poverty, unsupportive behaviors, or lack of knowledge weakened secondary appraisal, thereby intensifying stress and hindering coping.

Our findings reaffirm the critical role of family and social support in coping, while highlighting culturally specific mechanisms in the Iranian context. Participants emphasized the importance of spousal support and emotional, instrumental, and financial assistance from family, peers, and colleagues. In a study by Beckmann et al., participants highlighted the significance of financial and childcare support from family [33]. Additionally, emotional support from extended family and

fellow parents played a vital role in coping [34–42]. Importantly, in Iran, extended family members—motivated by strong cultural and religious expectations—often assume active caregiving roles, reducing both emotional and logistical burdens on parents, which contrasts with individualistic societies where nuclear family structures dominate caregiving. Similarly, while Pelletier et al. observed the value of connecting with other families who had undergone HSCT, our study shows that in Iran, such connections are facilitated through religious gatherings and community networks, enhancing emotional resilience [43]. Other studies indicated that flexible work arrangements and supportive employers help parents maintain employment during their child's HSCT [24,44]. In Iran, these arrangements are often informal or dependent on individual employer goodwill, reflecting systemic gaps in institutional support compared with high-income countries. Moreover, qualitative studies emphasized the importance of spiritual and emotional support from spouses, children, and close acquaintances [34,35,38,41,42,45–47]. In the Iranian context, communal prayers and religious rituals are deeply integrated into coping strategies, offering solace and facilitating access to broader community support. This culturally embedded spirituality highlights the need for culturally sensitive interventions that incorporate both emotional and spiritual support mechanisms.

Our findings revealed that some participants faced challenges due to inadequate family support and unsupportive behaviors from others. They reported insufficient spousal and financial support, bothersome behaviors from relatives and friends, and a lack of understanding from colleagues. While these themes align with previous studies, our research situates them within the Iranian sociocultural context, highlighting the absence of formal workplace mechanisms to support caregivers, which contrasts with high-income countries [44]. Participants relied on informal agreements or personal relationships with supervisors, underscoring systemic gaps in institutional support unique to Iran. Some Chinese fathers were found to delegate caregiving responsibilities primarily to mothers, offering less spousal support [41]. Additionally, parents often experienced neglect, disregard, and inconsistent support from friends and acquaintances, leading to frustration and resentment [37,48,49]. These findings emphasize the practical role of healthcare providers—especially nurses—in facilitating family communication, connecting parents to financial and social resources, and advocating for workplace accommodations, thereby helping parents cope with inadequate family and social support during HSCT.

Some participants expressed satisfaction with the support received from the healthcare system, including guidance, information, emotional support from the healthcare team, and expedited HSCT processes for their child. From the perspective of the Transactional Model, these interactions with the healthcare system provided parents with external resources that enhanced their secondary appraisal, increasing perceived coping capacity and reducing emotional distress. These findings align with previous studies showing that nurse education and active physician communication reassure parents and reduce anxiety [9,33,50,51]. Emotional support and meaningful interactions with healthcare providers helped families maintain hope, integrate care recommendations, and make informed decisions regarding HSCT [43,52]. Our study adds novel insights by situating these experiences within Iran's sociocultural and healthcare context, where structured psychosocial support systems are limited. Participants emphasized that expedited processes and personalized care by individual healthcare providers were crucial in mitigating logistical and emotional challenges, highlighting systemic gaps in formal support infrastructure and the central role of healthcare professionals as primary sources of psychosocial care.

The findings revealed significant gaps in support within the healthcare system. Participants reported challenges such as difficulties obtaining medications, limited treatment facilities, financial burdens, unsupportive behaviors from healthcare teams, inadequate information, and instances of neglect. Some parents expressed distrust when their preferences were ignored or explanations were insufficient [33], consistent with prior studies highlighting overlooked concerns and conflicting information from physicians [9,53]. This study provides unique insights into the sociocultural and systemic healthcare challenges faced by Iranian families during HSCT, including limited access to specialized pediatric care, financial constraints due to inadequate insurance, and regional treatment disparities. Unlike high-resource countries with structured support systems, Iranian families often rely on informal networks to fill gaps. Nurses and healthcare providers play a

crucial role in addressing these challenges through compassionate care, effective communication, and consistent support, highlighting actionable areas for policy and clinical improvement.

Participants emphasized the essential support provided by NGOs and religious groups during their children's HSCT journey. These external supports offered financial aid, spiritual assistance, and practical help, such as organizing ceremonies and activities for families, positively impacting caregivers' quality of life [41,54]. Studies also highlighted the role of churches and foundations in delivering comprehensive support, including meals and activities for healthy siblings [34,37,55]. Our study underscores the unique role of Iran's sociocultural framework, where NGOs and religious groups are central to community-based support, integrating spiritual and practical assistance in ways that differ from Western contexts. Families rely on religious organizations to meet emotional and spiritual needs, reflecting cultural emphasis on faith and communal solidarity. However, participants noted a decline in NGO support over time, consistent with other studies showing diminished social support after diagnosis [45,48].

Our study revealed that spiritual beliefs and attitudes toward transplantation play a key role in helping parents cope with their child's HSCT. Within the Transactional Model, these beliefs influenced both primary appraisal—by framing illness as a meaningful test rather than an uncontrollable threat—and secondary appraisal, by providing parents with perceived internal resources (faith, patience, acceptance) to manage the situation. Parents often view HSCT as a source of hope despite its risks, perceiving it as the best chance to treat their child and potentially extend life [43,56]. Similarly, many patients and caregivers see HSCT as a second chance at life, though some experience it as a frightening ordeal [9,57]. Spiritual beliefs also shape parents' understanding and acceptance of illness. Unlike Chinese parents, who often view cancer as "fate" [39,40], Iranian parents uniquely combine religious narratives with medical decisions, perceiving HSCT as both a medical intervention and a spiritual journey. This dual perspective highlights the importance for healthcare providers, especially nurses, to respect diverse belief systems and incorporate spiritual support into care, fostering a compassionate, holistic environment that enhances coping during HSCT.

Our findings indicated that factors such as financial stability, adequate insurance coverage, and the availability of a compatible donor facilitate coping during HSCT. A qualitative study found that having a family donor was fortunate, reducing complications and encouraging families to proceed [43], and the presence of an HLA-matched sibling donor alleviated procedural concerns [57]. This study extends existing research by highlighting challenges specific to Iranian families, including financial constraints, inadequate insurance, and limited access to compatible donors due to systemic healthcare limitations. Unlike countries with robust financial aid and donor registries, Iranian families face significant economic and infrastructural barriers. Cultural norms, including a strong sense of familial obligation, also shape donor recruitment and coping strategies. Nurses and healthcare providers should adopt culturally sensitive approaches, assisting families in navigating financial, insurance, and donor-related challenges to enhance coping and improve outcomes.

The findings identified several individual factors that hinder parents' ability to cope with their child's HSCT, including personality traits, job challenges, low economic status, lack of knowledge, and family issues. In terms of the Transactional Model, these barriers weakened parents' secondary appraisal by limiting available coping resources, thereby intensifying stress responses and reducing the likelihood of adaptive coping. Consistent with previous research, caregiving demands before and after HSCT often disrupt employment, causing financial stress [33,44,58,59]. Additional challenges include securing accommodation for follow-up care, income loss due to missed work, marital conflicts, and limited awareness of HSCT procedures and treatment options [24,41,57,60]. Prolonged hospitalization and follow-up care further exacerbate financial and emotional burdens [10,61–63]. This study extends existing knowledge by showing how these challenges are shaped by Iran's healthcare system and socio-economic context, including the lack of universal insurance and limited financial aid, which increase economic pressures compared with high-income countries. Cultural norms, such as traditional gender roles, often result in mothers shouldering the majority of caregiving responsibilities. To address these issues, comprehensive support is recommended, including full insurance coverage, flexible work arrangements, psychological counseling, and educational resources for parents navigating the HSCT process.

To operationalize the recommendation for comprehensive support, multi-level interventions are needed. At the health-care system level, multidisciplinary HSCT teams—including physicians, nurses, psychologists, and social workers—should coordinate care plans that address both medical and psychosocial needs of parents. Hospital-based support services, such as counseling units and patient navigators, can provide continuous guidance throughout the HSCT process. NGOs and charitable foundations can offer financial assistance, accommodation, and logistical support, while community-based networks can facilitate peer-to-peer emotional support. Integrating these resources into a cohesive, accessible framework can help ensure that parents receive timely, adequate, and holistic assistance.

### 4.1. Limitations

This study has several limitations. Its qualitative design emphasizes depth and subjective experiences rather than generalizability, which is inherent to this approach. Conducted at a single institution in Iran, the unique characteristics of the setting may have influenced participants' experiences, limiting transferability to hospitals or regions with different practices, resources, or demographics. The cross-sectional design also captures experiences at only one point in time. Furthermore, bereaved parents whose child had died during the HSCT process were excluded to avoid exacerbating grief, which may have limited the diversity of coping experiences represented. Although traditional forms of bias are not directly applicable in qualitative research, participants' perspectives may have been influenced by the interviewer, as well as by sociocultural or sociopolitical factors. Future research should address these limitations by recruiting a more diverse participant pool, conducting multi-center studies, and using longitudinal designs to explore how parents' coping strategies evolve over time.

## 5. Conclusions

This study highlights the crucial role of family, friends, and healthcare systems in supporting parents through their child's HSCT journey. Key facilitators include spousal support, workplace flexibility, financial aid, spiritual beliefs, and desirable individual situation while barriers such as inadequate family support, financial stress, and conflicts with healthcare teams also emerge. Nurses and healthcare providers are essential in bridging these gaps by offering emotional support, effective communication, and resource facilitation. External support from non-governmental organizations and religious groups significantly aids families, but challenges persist, underscoring the need for comprehensive and tailored support. Future research should explore specific subgroups, such as parents with limited social or financial resources, and employ longitudinal designs to examine how coping strategies evolve over time and to inform targeted interventions.

## Supporting information

**S1 File. The standards for reporting qualitative re-search (SRQR) guidelines.**
(DOCX)

## Acknowledgments

The authors express their heartfelt thanks to the participants who graciously dedicated their time and shared their invaluable experiences for this research. Additionally, we sincerely appreciate the financial support provided by Tehran University of Medical Sciences.

## Author contributions

**Conceptualization:** Maryam Maleki, Amir Ali Hamidieh, Abbas Mardani.
**Data curation:** Batool Pouraboli, Maryam Maleki, Nahid Dehghan Nayeri, Abbas Mardani.
**Formal analysis:** Batool Pouraboli, Maryam Maleki, Amir Ali Hamidieh.
**Investigation:** Nahid Dehghan Nayeri.

**Methodology:** Maryam Maleki, Nahid Dehghan Nayeri, Abbas Mardani.

**Supervision:** Batool Pouraboli, Nahid Dehghan Nayeri, Amir Ali Hamidieh.

**Validation:** Maryam Maleki, Amir Ali Hamidieh.

**Writing – original draft:** Batool Pouraboli, Maryam Maleki, Nahid Dehghan Nayeri, Amir Ali Hamidieh, Abbas Mardani.

**Writing – review & editing:** Maryam Maleki, Nahid Dehghan Nayeri, Abbas Mardani.

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
