## [Decision Letter · Decision Letter 0]

4 Jun 2025

Dear Dr. Maleki,

We look forward to receiving your revised manuscript.

Kind regards,

Ali Amanati

Academic Editor

PLOS ONE

Additional Editor Comments:

Dear Authors,‎

Your manuscript [PONE-D-25-19344] has passed the review stage and is ready for revision.‎

Editorial Comments:‎

To ensure the editor and reviewers can recommend that your revised manuscript be accepted, ‎please pay careful attention to each comment posted under this email. This approach will help us ‎avoid future clarifications and revisions, allowing us to move swiftly to a decision.‎

Technical points:‎

‎1. Please provide a point-by-point response to the Editor and reviewer's comments

‎2. Please highlight all the amends on your manuscript with a yellow color

‎3. Use line numbering and page number in the next submission‎

Reviewers' comments:

Reviewer's Responses to Questions

**Comments to the Author**

1. Is the manuscript technically sound, and do the data support the conclusions?

Reviewer #1: Yes

Reviewer #2: Yes

Reviewer #3: Yes

Reviewer #4: Partly

2. Has the statistical analysis been performed appropriately and rigorously?

Reviewer #1: Yes

Reviewer #2: N/A

Reviewer #3: Yes

Reviewer #4: N/A

3. Have the authors made all data underlying the findings in their manuscript fully available?

Reviewer #1: Yes

Reviewer #2: No

Reviewer #3: Yes

Reviewer #4: Yes

4. Is the manuscript presented in an intelligible fashion and written in standard English?

Reviewer #1: Yes

Reviewer #2: Yes

Reviewer #3: Yes

Reviewer #4: Yes

Reviewer #1: Reviewer Report

Manuscript Title: Obstacles and facilitators of parents' coping with hematopoietic stem cell transplantation of a child with cancer

Manuscript Number: PONE-D-25-19344

Title and Abstract

The title is clear and descriptive. The abstract succinctly captures the study's purpose, methodology, key findings, and implications. However, the term “oscillating support” may be ambiguous to international readers and could benefit from rephrasing to enhance clarity (e.g., “variable support”).

Introduction

The introduction is informative and lays a solid groundwork for the study. The use of global and national statistics supports the relevance of the topic. The theoretical anchoring in the Transactional Model of Stress and Coping is commendable. However, the narrative could be more concise by reducing repetitive content. A clearer articulation of the research gap early in the section would also strengthen the rationale.

Methods

The methodological rigor is evident in the thorough description of the study design, ethics, and trustworthiness strategies. The use of purposive sampling and adherence to SRQR enhance credibility. Nonetheless, further details on the recruitment process, especially how participant diversity was ensured, would improve the section’s transparency. A more explicit rationale for the sample size would also be helpful.

Results

The results are well-organized, with strong use of participant quotes to illustrate themes. The categorization into “oscillating support” and “beliefs and individual situation” effectively captures the complexity of parental coping. However, some subheadings could benefit from clearer wording. Streamlining some quotations may enhance readability without sacrificing depth.

Discussion

The discussion thoughtfully connects the findings to existing literature and highlights important cultural distinctions, particularly within the Iranian context. The comparison with other global studies adds value. However, there is occasional repetition of results, and the discussion could benefit from more emphasis on actionable implications. Stronger recommendations for policy or clinical practice would enhance its impact.

Limitations

The limitations are clearly acknowledged, including the single-center design and qualitative scope. Additional consideration of potential interviewer bias or sociopolitical influences on participant responses would provide a more comprehensive assessment.

Conclusions

The conclusion succinctly summarizes key findings and emphasizes the importance of holistic support systems. It could be strengthened by offering more specific directions for future research, such as targeting particular subgroups or employing longitudinal methods.

Overall Evaluation

This is a well-executed, culturally grounded qualitative study that offers meaningful insights into the experiences of parents navigating pediatric HSCT. With minor revisions to improve clarity, thematic labeling, and practical implications, the manuscript would be a strong candidate for publication in PLOS ONE.

Reviewer #2: This manuscript presents a timely and well-executed qualitative study that sheds light on Iranian parents’ lived experiences during their child’s haematopoietic stem cell transplantation (HSCT). The focus on both barriers and enablers of parental coping offers a balanced and nuanced contribution to the health social sciences literature. The writing is generally clear, and the integration of cultural context—particularly in the Discussion—is a commendable strength. The use of direct quotations enriches the narrative and gives voice to an often underrepresented population.

That said, some aspects of the manuscript would benefit from clarification and refinement. The methods section, while generally sound, needs clearer articulation of the analytical framework. Although the authors refer to conventional content analysis, their use of thematic language occasionally overlaps with approaches more typical of thematic analysis. It would strengthen the methodological rigor to either reaffirm their use of Graneheim & Lundman's content analysis or clarify if a hybrid approach was employed. Similarly, the manuscript would benefit from more explicit engagement with issues of reflexivity—how the researchers’ positions and assumptions were acknowledged and managed during data collection and analysis.

Sampling procedures and inclusion criteria are generally well described; however, it would be prudent to address how gatekeeping by nurses was mitigated and to reflect on the potential bias introduced by excluding bereaved parents. Data saturation is mentioned but not operationally defined; describing how and when saturation was determined (e.g. based on codebook stability or redundancy of themes) would enhance transparency. Also, while trustworthiness criteria are cited, concrete examples illustrating credibility, dependability, and confirmability would help ground these principles in practice.

The results section presents insightful findings, though the category structure could be refined. Some sub-categories conflate sources of support with the adequacy of that support, which may obscure the analytic clarity. The Discussion would benefit from reducing repetition of verbatim results and instead focusing on deeper interpretive synthesis. Additionally, the authors could specify which systems or actors might deliver the “comprehensive support” they recommend, to strengthen the paper’s applicability.

Finally, a few editorial adjustments—such as consistent tense usage, use of definite articles (e.g. “the largest Children’s Medical Center”), and spelling out abbreviations—would improve readability.

Reviewer #3: With the exception of replacing "1th" with "1st" and "2th" with "2nd", [Table 1], no changes are recommended. There is an orderly presentation of the results, which is a nicely detailed listing of important findings; the alternating description-quote for this evidence writing style is actually nicely 'to the point.' I might have expected a short listing or two interspersed here--but that probably doesn't add anything essential, and numbers of words limits to publications/length of article allow us to ignore that point. The value of applying qualitative research methodology for exploratory purposes, involving new therapeutics, is well illustrated here. The placement of barriers to change (patient/self, family/friends, associates/professionals, larger parts of society/culture) are well demonstrated here, and easy to review and make use of. There are certain underlying sociocultural aspects of this research subject and approach that make this article importantly definitive of the interculturalism, internationally shared qualities of life, aspects of health, and needs regarding health care, especially for conditions that are equally important across all disciplines and people. Sometimes, we relate quality of life and health care to the kind of society or country health care is taking place within, with the added impression that quality of care can differ greatly across different cultures and countries. But this article clearly demonstrates that certain human features about QOL and health care, that exist regarding of a culture or country's place in the world. This article puts that concept back into the conscious mindset of its readers, and other nations' healthcare givers. It reminds us that regardless of quality of care, availability of care, cost of care, most health matters are global, and most nations, cultures, need to be acknowledged for the efforts being made in handling their highest risk patients. This article is a very nice, detailed yet succinct example that I would share with my students, in my qualitative research methods course teachings.

Reviewer #4: You have taken on an important and emotionally complex topic by exploring the experiences of parents whose children undergo hematopoietic stem cell transplantation. I understand the inherent difficulties of conducting in-depth qualitative research with this vulnerable population, and I appreciate your effort. I provide several helpful recommendations below to improve your manuscript's theoretical alignment, methodological clarity, and transparency.

1. Reflexivity is not adequately addressed in the manuscript. Important details about the research team are absent, such as:

Which author or authors conducted the interviews?

What qualifications and qualitative methods training did they have?

Which gender were they?

Before the interviews, were relationships built with the participants?

This information is required to assess possible influences on data collection and interpretation. In the Methods section, I suggest adding a succinct paragraph describing the interviewers' backgrounds and any connections to participants.

2. Several important study design elements are not disclosed:

How participants were approached (e.g., face-to-face, telephone, etc.) is unclear.

There was no mention of whether any individuals refused to participate or dropped out or whether refusals were treated as exclusion criteria.

It is also not stated whether field notes were made during or after the interviews.

3- The description of the data analysis process lacks several critical details:

How many researchers were involved in coding the data?

What software was used to manage the data?

Were the themes derived inductively or identified in advance?

Did participants review the findings (member checking)?

How were discrepancies between coders resolved?

How did you use triangulation strategies to investigate data validity?

4. Despite claiming in the introduction that the study is grounded inn Lazarus and Folkman's Transactional Model, the methodology, analysis, or discussion do not significantly incorporate the model. The interview guide, coding framework, or thematic analysis have not been organized according to the model's central concepts (primary and secondary appraisal, for example). I suggest specifically integrating the model into the analytical and discussion sections to guarantee theoretical consistency.

**Do you want your identity to be public for this peer review?** For information about this choice, including consent withdrawal, please see our Privacy Policy

Reviewer #1: **Yes: ** Juliana Aggrey

Reviewer #2: **Yes: ** Alireza abbasi

Reviewer #3: **Yes: ** Brian L Altonen

Reviewer #4: No

---

## [Author Response · Author response to Decision Letter 1]

15 Aug 2025

Dear Editor, PLOS ONE

Thank you for this opportunity to revise and resubmit our article. We sincerely thank the reviewers for their comments and the provision of constructive comments to improve the quality of our article. Accordingly, text modifications were made, which have been detailed point-by-point below. All changes were highlighted throughout the manuscript using red color.

Comments by reviewer 1

Title and Abstract

1. The title is clear and descriptive. The abstract succinctly captures the study's purpose, methodology, key findings, and implications. However, the term “oscillating support” may be ambiguous to international readers and could benefit from rephrasing to enhance clarity (e.g., “variable support”).

Answer: We appreciate the reviewer’s insightful comment. To improve clarity for international readers, we have replaced the term “oscillating support” with “variable support” throughout the manuscript, including in the abstract, results, discussion, Table 2, and Figure.

Introduction

2. The introduction is informative and lays a solid groundwork for the study. The use of global and national statistics supports the relevance of the topic. The theoretical anchoring in the Transactional Model of Stress and Coping is commendable. However, the narrative could be more concise by reducing repetitive content. A clearer articulation of the research gap early in the section would also strengthen the rationale.

Answer: We thank the reviewer for the helpful suggestions. In response, we revised the introduction to reduce repetitive content and improve clarity while maintaining essential details. We also moved the articulation of the research gap to an earlier point in the section, emphasizing -the lack of qualitative research on parental coping in the Iranian context (Line: 47-84).

Methods

3. The methodological rigor is evident in the thorough description of the study design, ethics, and trustworthiness strategies. The use of purposive sampling and adherence to SRQR enhance credibility. Nonetheless, further details on the recruitment process, especially how participant diversity was ensured, would improve the section’s transparency. A more explicit rationale for the sample size would also be helpful.

Answer: To address this comment, we expanded the data collection section to provide more detail on the recruitment process and how participant diversity was ensured. Specifically, we clarified the steps taken to recruit participants from varied demographic and clinical backgrounds, including differences in gender, socioeconomic status, geographic location, and type of HSCT. We also added an explicit rationale for the sample size, explaining that the final number of 20 participants was guided by the principle of data saturation and is consistent with qualitative research standards for in-depth exploration (Line: 106-121; 132-137).

Results

4. The results are well-organized, with strong use of participant quotes to illustrate themes. The categorization into “oscillating support” and “beliefs and individual situation” effectively captures the complexity of parental coping. However, some subheadings could benefit from clearer wording. Streamlining some quotations may enhance readability without sacrificing depth.

Answer: We thank the reviewer for these helpful suggestions. In response, we have revised some subheadings in the Results section to improve clarity for international readers. Specifically, “Oscillating support” has been replaced with “Variable support,” and other subheadings have been reworded for greater precision (e.g., “Inadequate support from family and friends” to “Insufficient support from family and friends,” “Unfavorable support from the healthcare system” to “Challenges in support from the healthcare system,” and “Obstructive individual situation” to “Barriers related to individual circumstances”). We also streamlined several participant quotations by removing redundant details while preserving their original meaning and emotional impact, thereby enhancing readability without reducing depth.

Discussion

5. The discussion thoughtfully connects the findings to existing literature and highlights important cultural distinctions, particularly within the Iranian context. The comparison with other global studies adds value. However, there is occasional repetition of results, and the discussion could benefit from more emphasis on actionable implications. Stronger recommendations for policy or clinical practice would enhance its impact.

Answer: We thank the reviewer for the valuable comment. In response, we have revised the discussion to reduce repetition, clarify key findings, and emphasize actionable implications for policy and clinical practice. We highlighted the unique Iranian sociocultural and religious context in family and community support, identified systemic gaps in healthcare such as limited insurance and informal workplace support, and emphasized the critical roles of nurses and healthcare providers in facilitating communication, providing emotional and financial guidance, and supporting workplace accommodations. Additionally, we added clear recommendations, including comprehensive insurance coverage, flexible work arrangements, psychological counseling, and educational resources for parents navigating HSCT (Line: 384-513).

Limitations

6. The limitations are clearly acknowledged, including the single-center design and qualitative scope. Additional consideration of potential interviewer bias or sociopolitical influences on participant responses would provide a more comprehensive assessment.

Answer: We have revised the limitations section to clarify that, as a qualitative study, the findings reflect subjective experiences rather than generalizable results, and although traditional bias is not applicable, participants’ perspectives may be influenced by interviewer perspective, sociocultural or sociopolitical factors. We have highlighted the single-center and cross-sectional design limitations and suggested future multi-center and longitudinal research to enhance transferability and capture evolving coping strategies (Line: 515-526).

Conclusions

7. The conclusion succinctly summarizes key findings and emphasizes the importance of holistic support systems. It could be strengthened by offering more specific directions for future research, such as targeting particular subgroups or employing longitudinal methods.

Answer: We have revised the conclusion to include more specific directions for future research, highlighting the need to study particular subgroups, such as parents with limited social or financial resources, and to employ longitudinal designs to capture the evolution of coping strategies (Line: 535-538).

Comments by reviewer 2

This manuscript presents a timely and well-executed qualitative study that sheds light on Iranian parents’ lived experiences during their child’s haematopoietic stem cell transplantation (HSCT). The focus on both barriers and enablers of parental coping offers a balanced and nuanced contribution to the health social sciences literature. The writing is generally clear, and the integration of cultural context—particularly in the Discussion—is a commendable strength. The use of direct quotations enriches the narrative and gives voice to an often underrepresented population.

1. That said, some aspects of the manuscript would benefit from clarification and refinement. The methods section, while generally sound, needs clearer articulation of the analytical framework. Although the authors refer to conventional content analysis, their use of thematic language occasionally overlaps with approaches more typical of thematic analysis. It would strengthen the methodological rigor to either reaffirm their use of Graneheim & Lundman's content analysis or clarify if a hybrid approach was employed.

Answer: We appreciate the reviewer’s insightful observation. We confirm that the study employed conventional content analysis as described by Graneheim and Lundman, and not a hybrid thematic approach. To avoid ambiguity, we revised the description of the analytical process to clearly state that data were coded and categorized inductively in alignment with this method (Line: 151-160).

2. Similarly, the manuscript would benefit from more explicit engagement with issues of reflexivity—how the researchers’ positions and assumptions were acknowledged and managed during data collection and analysis.

Answer: We appreciate the reviewer’s comment on reflexivity. In response, we added a dedicated paragraph in the Methods section describing how the researchers’ positions and assumptions were acknowledged and managed during data collection and analysis. This includes documentation of the interviewer’s professional background, use of reflexive notes, and ongoing team discussions to mitigate bias and ensure that findings reflect participants’ perspectives (Line: 138-149).

3. Sampling procedures and inclusion criteria are generally well described; however, it would be prudent to address how gatekeeping by nurses was mitigated and to reflect on the potential bias introduced by excluding bereaved parents. Data saturation is mentioned but not operationally defined; describing how and when saturation was determined (e.g. based on codebook stability or redundancy of themes) would enhance transparency. Also, while trustworthiness criteria are cited, concrete examples illustrating credibility, dependability, and confirmability would help ground these principles in practice.

Answer: To address this comment, we have taken the following steps to enhance methodological clarity and transparency: (1) We clarified how gatekeeping by nurses was minimized by instructing them to introduce the study broadly to all eligible parents without pre-screening, followed by independent review of medical records by the research team (Line: 106-114); (2) We provided an operational definition of data saturation as the point when no new codes, categories, or themes emerged in three consecutive interviews and the codebook demonstrated stability (Line: 132-137); (3) We expanded the Trustworthiness section to include concrete examples of how credibility, dependability, and confirmability were achieved in practice, such as member checking, peer validation, triangulation, and maintaining an audit trail (Line: 162-176); (4) In the Limitations section, we acknowledged the exclusion of bereaved parents as a potential source of bias that may have limited the diversity of coping experiences captured in the study (Line: 515-526).

4. The results section presents insightful findings, though the category structure could be refined. Some sub-categories conflate sources of support with the adequacy of that support, which may obscure the analytic clarity. The Discussion would benefit from reducing repetition of verbatim results and instead focusing on deeper interpretive synthesis. Additionally, the authors could specify which systems or actors might deliver the “comprehensive support” they recommend, to strengthen the paper’s applicability.

Answer: We appreciate the reviewer’s insightful comments on the Results section. In revising the manuscript, we refined the wording of certain category and subcategory titles to improve clarity and ensure that their focus is easily understood. We also streamlined some of the participant quotations by removing non-essential details while preserving their original meaning, thus enhancing readability without losing depth. While we acknowledge the potential to further separate “sources of support” from “adequacy of support”, we decided to retain the overall category structure because it reflects the way participants themselves intertwined these aspects in their narratives. Many participants described the source and adequacy of support in a single account, making it analytically meaningful to present them together. We believe that maintaining this integrated structure preserves the authenticity of participants’ experiences and remains consistent with the inductive nature of our conventional content analysis (Result section).

In the Discussion, we have reduced repetition of verbatim participant quotations and placed more emphasis on interpretive synthesis, linking our findings to existing literature and theoretical frameworks. Additionally, as recommended, we have added a concluding paragraph specifying which systems and actors—such as multidisciplinary HSCT teams, hospital-based counseling services, non-governmental organizations, and community support networks—could deliver the comprehensive support identified as necessary in our study (Line: 505-513).

5. Finally, a few editorial adjustments—such as consistent tense usage, use of definite articles (e.g. “the largest Children’s Medical Center”), and spelling out abbreviations—would improve readability.

Answer: We thank the reviewer for the editorial suggestions. We have carefully standardized tense usage, ensured correct use of definite articles, and spelled out all abbreviations at their first mention throughout the manuscript to improve clarity and readability.

Comments by reviewer 3

With the exception of replacing "1th" with "1st" and "2th" with "2nd", [Table 1], no changes are recommended. There is an orderly presentation of the results, which is a nicely detailed listing of important findings; the alternating description-quote for this evidence writing style is actually nicely 'to the point.' I might have expected a short listing or two interspersed here--but that probably doesn't add anything essential, and numbers of words limits to publications/length of article allow us to ignore that point. The value of applying qualitative research methodology for exploratory purposes, involving new therapeutics, is well illustrated here. The placement of barriers to change (patient/self, family/friends, associates/professionals, larger parts of society/culture) are well demonstrated here, and easy to review and make use of. There are certain underlying sociocultural aspects of this research subject and approach that make this article importantly definitive of the interculturalism, internationally shared qualities of life, aspects of health, and needs regarding health care, especially for conditions that are equally important across all disciplines and people. Sometimes, we relate quality of life and health care to the kind of society or country health care is taking place within, with the added impression that quality of care can differ greatly across different cultures and countries. But this article clearly demonstrates that certain human features about QOL and health care, that exist regarding of a culture or country's place in the world. This article puts that concept back into the conscious mindset of its readers, and other nations' healthcare givers. It reminds us that regardless of quality of care, availability of care, cost of care, most health matters are global, and most nations, cultures, need to be acknowledged for the efforts being made in handling their highest risk patients. This article is a very nice, detailed yet succinct example that I would share with my students, in my qualitative research methods course teachings.

Answer: We sincerely thank the reviewer for the thoughtful and encouraging feedback. We greatly appreciate your recognition of the clarity, organization, and level of detail in our results presentation, as well as your acknowledgment of the value of applying qualitative methodology to this research topic. We are also grateful for your insightful reflections on the intercultural and globally relevant aspects of quality of life and health care illustrated in our findings.

As suggested, we have corrected “1th” to “1st” and “2th” to “2nd” in Table 1. We are encouraged by your remarks that the manuscript provides a succinct yet detailed example of qualitative research that could be useful in teaching contexts. Your positive evaluation reinforces the importance of the study’s contribution and motivates us to further pursue research on this topic.

Comments by reviewer 4

You have taken on an important and emotionally complex topic by exploring the experiences of parents whose children undergo hematopoietic stem cell transplantation. I understand the inherent

---

## [Decision Letter · Decision Letter 1]

14 Sep 2025

Dear Dr. Maleki,

Thank you for submitting your manuscript to PLOS ONE. After careful consideration, we feel that it has merit but does not fully meet PLOS ONE’s publication criteria as it currently stands. Therefore, we invite you to submit a revised version of the manuscript that addresses the points raised during the review process. Please submit your revised manuscript by 2 weeks. If you will need more time than this to complete your revisions, please reply to this message or contact the journal office at plosone@plos.org . A rebuttal letter that responds to each point raised by the academic editor and reviewer(s). You should upload this letter as a separate file labeled 'Response to Reviewers'.A marked-up copy of your manuscript that highlights changes made to the original version. You should upload this as a separate file labeled 'Revised Manuscript with Track Changes'.An unmarked version of your revised paper without tracked changes. You should upload this as a separate file labeled 'Manuscript'.

We look forward to receiving your revised manuscript.

Kind regards,

*
**Ali Amanati**
*

**Academic Editor**

PLOS ONE

Journal Requirements:

Additional Editor Comments:

Dear authors,

‎The invited reviewer (Reviewer #4) posted new comments. So, the manuscripts ‎require a ‎round of revision.‎ Please provide a point-by-point response to the ‎reviewer's ‎comments and highlight all the ‎amends on your manuscript with ‎yellow color.‎

Yours,

Reviewers' comments:

Reviewer's Responses to Questions

**Comments to the Author**

Reviewer #1: All comments have been addressed

Reviewer #2: All comments have been addressed

Reviewer #3: All comments have been addressed

Reviewer #4: (No Response)

2. Is the manuscript technically sound, and do the data support the conclusions?

Reviewer #1: Yes

Reviewer #2: Yes

Reviewer #3: Yes

Reviewer #4: Yes

3. Has the statistical analysis been performed appropriately and rigorously?

Reviewer #1: Yes

Reviewer #2: Yes

Reviewer #3: Yes

Reviewer #4: N/A

4. Have the authors made all data underlying the findings in their manuscript fully available?

Reviewer #1: Yes

Reviewer #2: No

Reviewer #3: Yes

Reviewer #4: Yes

5. Is the manuscript presented in an intelligible fashion and written in standard English?

Reviewer #1: Yes

Reviewer #2: Yes

Reviewer #3: Yes

Reviewer #4: Yes

Reviewer #1: Based on my re-review of the manuscript, "Obstacles and facilitators of parents' coping with hematopoietic stem cell transplantation of a child with cancer," I have concluded that the authors have successfully addressed the concerns raised by the reviewers. The revisions have significantly improved the manuscript, and it is now suitable for publication.

Here is a summary of the key revisions and how they have improved the manuscript:

• Clarity and Readability: The authors have revised the title and abstract, replacing the ambiguous term "oscillating support" with "variable support" to improve clarity for international readers. They also reworded several subheadings in the results section for greater precision.

• Methodological Rigor and Transparency: The authors have provided a more explicit rationale for their sample size, explaining that the final number of 20 participants was guided by data saturation. They also clarified the recruitment process to ensure participant diversity. Furthermore, they added a dedicated paragraph on reflexivity, detailing how the researchers' positions and assumptions were managed during data collection and analysis. The manuscript also now includes an operational definition of data saturation and concrete examples of how trustworthiness criteria were met, such as member checking and peer validation.

• Theoretical Consistency: While the study used a conventional content analysis method, the authors have now integrated an explicit interpretation in the discussion section to link their inductively derived categories to the key concepts of the Transactional Model of Stress and Coping, strengthening the paper's theoretical consistency.

• Actionable Implications: The discussion has been revised to reduce repetition and place more emphasis on actionable implications for policy and clinical practice. The authors added clear recommendations for comprehensive support systems, including roles for multidisciplinary teams, hospital-based counseling, and community networks.

• Limitations and Bias: The authors have acknowledged the potential bias introduced by excluding bereaved parents from the study and noted that the findings are specific to a single institution in Iran, limiting their generalizability. They also added a note on potential influences from the interviewer and socio-political factors.

The authors' thorough and well-reasoned responses demonstrate their commitment to improving the manuscript. The revisions have not only addressed the specific points raised by the reviewers but have also strengthened the overall quality and contribution of the paper.

Reviewer #2: I appreciate the authors' response and the integration of the Transactional Model of Stress and Coping into the Discussion section, as this adequately addresses my previous comment. However, it would have been beneficial if the authors had demonstrated more explicitly how the concepts of primary and secondary appraisal influenced the coding or theme generation. Since these concepts are central to the theoretical framework, further clarification of their role in shaping the analysis would strengthen the theoretical consistency and transparency of the study's methodology.

Reviewer #3: Thank you for sharing much more on your research and analysis methods and how the tasks for each individual were managed. The narratives/quotes add substantially to the body and value this article, in terms of utilizing it as an example to my colleagues and co-researchers on how to engage in value-driven quality of care research and related QI programs. Understanding the goals for this research, reading in detail how patients, patient's families and care givers are impacted by these processes, hopefully enables these program to grow more rapidly and effectively at the patient and institutional health education level.

Reviewer #4: Thank you for the response. However, I was unable to locate the claimed integration of the Transactional Model in the Discussion section. The manuscript still lacks a meaningful theoretical framework that connects findings to broader analytical concepts. Without this, the study remains descriptive and does not move beyond context-specific observations. Theory provides the necessary link between observing something and giving it meaning.

**Do you want your identity to be public for this peer review?** For information about this choice, including consent withdrawal, please see our Privacy Policy

Reviewer #1: **Yes: ** Juliana Aggrey

Reviewer #2: **Yes: ** Alireza Abbasi

Reviewer #3: **Yes: ** Brian L Altonen

Reviewer #4: No

---

## [Author Response · Author response to Decision Letter 2]

16 Sep 2025

Dear Editor, PLOS ONE

Thank you for this opportunity to revise and resubmit our article. We would like to express our sincere gratitude to all the reviewers for their thoughtful feedback. We especially thank the two reviewers who endorsed our manuscript for publication, as well as the reviewer who provided detailed suggestions for revision. Their constructive comments were invaluable in improving the clarity, theoretical integration, and overall quality of our work.

Accordingly, some text modifications were made and are presented point-by-point below. All changes have been highlighted throughout the manuscript using red color.

Sincerely Yours

Comments by reviewer 2

I appreciate the authors' response and the integration of the Transactional Model of Stress and Coping into the Discussion section, as this adequately addresses my previous comment. However, it would have been beneficial if the authors had demonstrated more explicitly how the concepts of primary and secondary appraisal influenced the coding or theme generation. Since these concepts are central to the theoretical framework, further clarification of their role in shaping the analysis would strengthen the theoretical consistency and transparency of the study's methodology.

Answer: Thank you very much for your constructive feedback. We would like to clarify that our study employed a conventional content analysis approach, rather than a directed content analysis. In conventional content analysis, categories are inductively generated from the data without being constrained by a predefined theoretical framework. Therefore, the coding and theme development in our study were grounded in participants’ narratives rather than explicitly guided by the concepts of primary and secondary appraisal.

However, in line with your valuable suggestion, we sought to connect our findings to the Transactional Model of Stress and Coping at the stage of interpretation and discussion. By doing so, we aimed to demonstrate how the parents’ experiences—such as fear, uncertainty, financial strain, and reliance on family and spiritual support—can be understood through the processes of primary and secondary appraisal. This approach allowed us to preserve the inductive integrity of conventional content analysis, while still situating our findings within a meaningful theoretical framework (Line: 381-388; 427-430; 466-469; 493-495).

Comments by reviewer 4

Thank you for the response. However, I was unable to locate the claimed integration of the Transactional Model in the Discussion section. The manuscript still lacks a meaningful theoretical framework that connects findings to broader analytical concepts. Without this, the study remains descriptive and does not move beyond context-specific observations. Theory provides the necessary link between observing something and giving it meaning.

Answer: We sincerely thank you for your valuable comment. In response, we have revised the Discussion section to more explicitly integrate the Transactional Model of Stress and Coping as the theoretical framework. Specifically, we now interpret our findings in terms of primary appraisal (e.g., parents’ reported fear, financial stress, and conflict with healthcare teams, which reflect the perception of HSCT as a highly stressful and threatening situation) and secondary appraisal (e.g., parents’ references to spousal support, NGO assistance, financial stability, and spiritual beliefs, which represent resources that helped them to cope). We further highlighted that when resources were perceived as sufficient, parents described resilience and hope, while lack of resources undermined coping and intensified stress (Line: 381-388; 427-430; 466-469; 493-495).

We hope that the changes meet your journal needs.

Sincerely Yours,

Authors

---

## [Decision Letter · Decision Letter 2]

7 Oct 2025

Obstacles and facilitators of parents' coping with hematopoietic stem cell transplantation of a child with cancer

PONE-D-25-19344R2

Dear Dr. Maryam Maleki,

We’re pleased to inform you that your manuscript has been judged scientifically suitable for publication and will be formally accepted for publication once it meets all outstanding technical requirements.

Kind regards,

*
**Ali Amanati**
*

**Academic Editor**

PLOS ONE

Additional Editor Comments (optional):

The authors have effectively utilized all available resources and data to enhance the manuscript, making it more scientifically robust than before. Therefore, based on my opinion and the esteemed ‎reviewers' ‎comments, it could be published in its current form.‎

Yours‎,

Reviewers' comments:

Reviewer's Responses to Questions

**Comments to the Author**

Reviewer #4: All comments have been addressed

2. Is the manuscript technically sound, and do the data support the conclusions?

Reviewer #4: Yes

3. Has the statistical analysis been performed appropriately and rigorously?

Reviewer #4: N/A

4. Have the authors made all data underlying the findings in their manuscript fully available?

Reviewer #4: Yes

5. Is the manuscript presented in an intelligible fashion and written in standard English?

Reviewer #4: Yes

Reviewer #4: (No Response)

**Do you want your identity to be public for this peer review?** For information about this choice, including consent withdrawal, please see our Privacy Policy

Reviewer #4: No

---

## [Editor Report · Acceptance letter]

PONE-D-25-19344R2

PLOS ONE

Dear Dr. Maleki,

I'm pleased to inform you that your manuscript has been deemed suitable for publication in PLOS ONE. Congratulations! Your manuscript is now being handed over to our production team.

Kind regards,

on behalf of

Professor Ali Amanati

Academic Editor

PLOS ONE